# Surgical and transcatheter aortic valve replacement after orthotopic heart transplantation: a case series

Luise Roehrich [1,2,3,8] ✉, Hristian Hinkov [1,2,3,4,8], Christoph Knosalla [1,2,3], Felix Hennig[1,2,3], Christoph Klein[2,4,5], Marian Kukucka[2,4,6], Nicolas Merke[1,2], Laurenz Kopp Fernandes[1,2], Isabell Anna Just[1,2,3], Sascha Ott[1,2,3,6], Henryk Dreger [2,3,4,5], Volkmar Falk [1,2,3,7], Axel Unbehaun[1,2,3,4,9] & Felix Schoenrath[1,2,3,9]

## Abstract

**Background** Aortic valve dysfunction is a rare but relevant long-term complication after orthotopic heart transplantation (HTX). Treatment options include surgical (SAVR) and transcatheter (TAVR) aortic valve replacement, but evidence is limited to the level of case reports.

**Methods** A total of 2054 patients underwent HTX between 1986 and 2023 at the Deutsches Herzzentrum der Charité, Berlin, Germany, 16 of them underwent aortic valve replacement (SAVR, N = 7; TAVR, N = 9) after HTX. In this case series we report on the outcomes of these 16 patients (age 29-73 years).

**Results** Isolated aortic valve regurgitation occurs in 5 (31%) patients (TAVR N = 1, SAVR N = 4), while 11 patients (69%) suffer from aortic valve stenosis. Severe pre-procedural heart failure is present in 38% of the patients. Thirty-day mortality is 0%, in-hospital mortality is N = 2 (22%) patients due to sepsis, both patients were severely decompensated prior to TAVR. An uneventful postoperative course occurs in 8 (50%) patients, the patient's functional status improves in 12 (75%) cases.

**Conclusion** TAVR is the increasingly preferred treatment for aortic valve dysfunction after heart transplantation, but SAVR is a feasible alternative for individuals who are ineligible for TAVR. Further information is needed on the appropriate procedural timing in these high-risk patients.

## Plain Language Summary

Developing aortic valve disease, which comes from stiffness and/or leakage of the valve, is rare after heart transplantation, but leads to recurrence of heart failure with a high burden of symptoms. The treatment in severe cases is the replacement of the dysfunctional valve either by surgery or by using a thin tube, which is inserted through the blood vessels (catheter). Here we share our center's experience with aortic valve procedures after heart transplantation in 16 patients. Valve replacement via catheter is preferred as it limits trauma for the patient, but surgical replacement is feasible in individuals who are ineligible for the catheter procedure. There are special concerns for redo-procedure with the risk of periprocedural infections; however, 75% of our patients showed an improvement in their well-being and quality of life after treatment.

Heart transplantation is the therapeutic gold standard for patients with end-stage heart failure, with a median survival time of 11 years[1]. Contemporary surgical techniques, immunosuppressive medication, improved graft preservation techniques, as well as management of life-threatening complications and comorbidities, have improved long-term survival after orthotopic heart transplantation[2]. Long-term complications, such as degenerative valvular heart diseases, become more frequent with improved longevity of the allograft. Secondary valvular dysfunction is rare with an estimated prevalence of 6.6% being diagnosed on the average 11.31 ± 6.95 years after transplantation[3]. Eighty-nine percent of these patients were treated conservatively, but severe valvular dysfunction can lead to recurrent heart failure and affects the quality of life of our patients, claiming surgical or interventional treatment[3]. The presentation and the etiology of structural heart valve deterioration differ from that in native hearts: while degenerative

[1]Deutsches Herzzentrum der Charité (DHZC), Department of Cardiothoracic and Vascular Surgery, Augustenburger Platz 1, Berlin, Germany. [2]Charité-Universitätsmedizin Berlin, corporate member of Freie Universität Berlin and Humboldt-Universität zu Berlin, Charitéplatz 1, Berlin, Germany. [3]DZHK (German Center for Cardiovascular Research), partner site Berlin, Berlin, Germany. [4]Deutsches Herzzentrum der Charité (DHZC), Structural heart interventions program (SHIP), Augustenburger Platz 1, Berlin, Germany. [5]Deutsches Herzzentrum der Charité (DHZC), Department of Cardiology, Angiology and Intensive Care Medicine, Augustenburger Platz, Berlin, Germany. [6]Deutsches Herzzentrum der Charité, Department of Cardiac Anaesthesiology and Intensive Care Medicine, Berlin, Germany. [7]ETH Zurich, Dept. Health Sciences and Technology, Translational Cardiovascular Technology, Zurich, Switzerland. [8]These authors contributed equally: Luise Roehrich, Hristian Hinkov. [9]These authors jointly supervised this work: Axel Unbehaun, Felix Schoenrath. ✉e-mail: luise.roehrich@dhzc-charite.de

**Fig. 1 | Time-dependent distribution of SAVR and TAVR between 1998 and 2024.** Time-dependent display of SAVR and TAVR according to their indication. SAVR = surgical aortic valve replacement; TAVR = transcatheter aortic valve replacement.

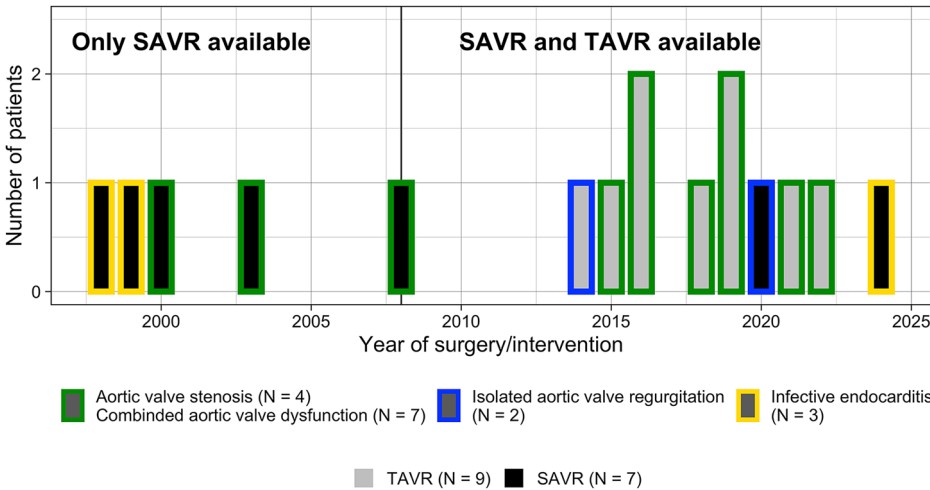

aortic stenosis is the most common valvular dysfunction, followed by mitral regurgitation in native hearts, in cardiac allografts tricuspid (94%), mitral (2.6%) and aortic (1.7%) valve regurgitation are the most common valvular pathologies, followed by aortic stenosis (1.3%) and mitral stenosis (0.35%)[3,4].

Transcatheter aortic valve replacement (TAVR) is an established therapy in all risk categories of patients, and recent data show equipoise in outcome compared with surgical aortic valve replacement (SAVR)[5,6]. Although not explicitly excluded, heart transplanted patients were not included in the DEDICATE or PARTNER trials, and the NOTION trial excluded patients with prior heart surgery[5–7]. The evidence for the treatment of secondary aortic valve insufficiency or stenosis in a transplanted heart is limited to anecdotal case reports and one small series of 7 patients (5 SAVR and 2 TAVR). Here, we describe our single-center experience in a high-volume transplantation center with surgical and interventional treatment of aortic valve dysfunction after heart transplantation.

Our data confirm, that aortic valve dysfunction is a rare, but relevant long-term complication after heart transplantation. We find that TAVR is currently the preferred approach for treatment of aortic valve dysfunction after heart transplantation in our center, but SAVR is a feasible alternative in individuals who are ineligible for TAVR. The peri-procedural risks are higher than in non-transplant patients. However, 75% of the patients report a subjective improvement in their well-being and functional status after the procedure.

## Methods
### Study population
A retrospective screening was conducted on all medical records of 2054 patients who underwent orthotopic heart transplantation between 1986 and 2023 at the Deutsches Herzzentrum der Charité, Berlin, Germany. Sixteen adult patients of these underwent surgical or interventional aortic valve replacement after HTX and met the inclusion criteria; therefore, they were included in this case series. Pediatric cases were excluded, as were all concomitant valve procedures during orthotopic heart transplantation or within one year of the transplantation. One patient underwent TAVR after heart transplantation, and in the following course, a redo-TAVR valve-in-valve procedure after 9 years: in this case, data from the first procedure were described in this analysis.

Prior to 2008, when the first TAVR was performed in our center, SAVR was the only therapeutic option for severe aortic valve dysfunction. Six years after the introduction of TAVR (2014), the first heart transplanted patient underwent TAVR and TAVR has been increasingly used since then. The development between 1998 and 2024 is displayed in Fig. 1: Time-dependent distribution of SAVR and TAVR between 1998 and 2024.

The choice of approach is an interdisciplinary, individualized heart team decision with regard to the underlying anatomy and pathology,

suitable landing zone and access for TAVR, and relevant comorbidities (Fig. 2: Heart team decision tree)

The ethics committee of Charité – Universitätsmedizin Berlin (Ethikausschuss am Campus Mitte) granted approval and waived patients' consent due to the retrospective, anonymized nature of this analysis (EA1/255/24).

### Data collection
The data was collected manually from the electronic patient management system. The pre-procedural data set comprises demographic specifics, details concerning the heart transplantation, immunosuppressive therapy and specific comorbidities as well as laboratory and clinical details of the patient. The functional status of the patient is represented by New York Heart Association (NYHA) Classification. Pre-procedural decompensation was defined by the presence of one or more of the following signs: pleural effusion, ascites, new or progressive peripheral edema, worsening of dyspnea to NYHA III or IV, pulmonary edema with or without consecutive pneumonia or new-onset dialysis due to acute (worsening of) cardio-renal syndrome. The amount of vasoactive therapy is stated as vasoactive inotropic score (VIS)[8].

We provide information about the technical details of the intervention/surgery. Outcome measures include intra-hospital, 30-day, early (> 30 days and ≤ 1 year) and late (> 1 year and ≤3 years) mortality[9]. Additionally, major adverse events include stroke, sepsis, new onset dialysis, readmission, need for pacemaker implantation and access site complications according to Valve Academic Research Consortium (VARC)−3 criteria[9]. Further outcome indicators include changes in NYHA class, which serves as a surrogate for the functional status, echocardiographic findings including paravalvular leakage, mean pressure gradient and left ventricular ejection fraction.

Transplant specific data was missing for one TAVR patient and can be found for the others in Supplementary Data 1. The supplementary material also includes a short case report of each patient (Supplementary Data 2). Echocardiographic specifics of the aortic valve are included in Table 2 and Supplementary Data 3.

### Statistics and reproducibility
Categorial data is displayed in numbers and percent, continuous data is displayed as median and interquartile range (IQR). Survival data is displayed as Kaplan-Meier-Analysis to help visualizing the survival, however group comparison was not feasible due to the limited number of patients. All data should be read descriptively. RStudio version 2022.12.0 + 353 (R version 4.2.2.) was used for data analysis with the following packages: "base" (version 4.2.2), "rio" (version 1.1.1), "survival" (version 3.5-0), "survminer" (version 0.4.9), "ggplot2" (version 3.4.0), "ggpubr" (version 0.5.0).

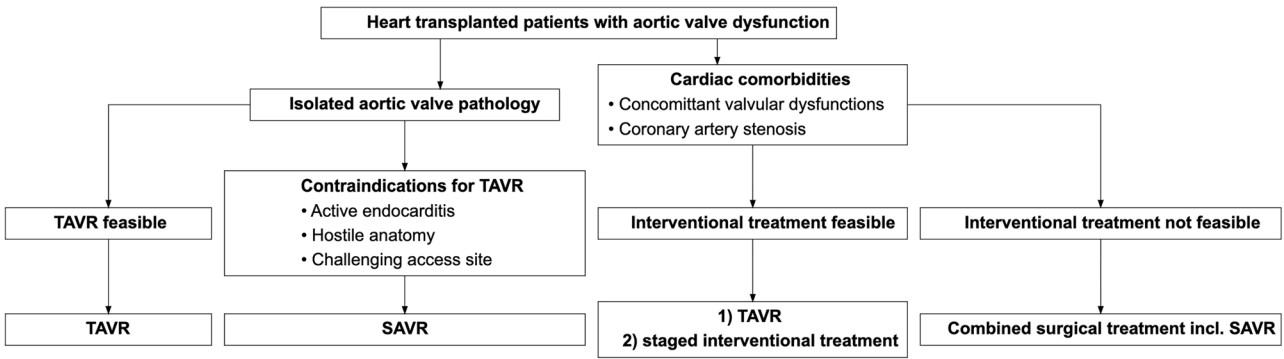

**Fig. 2 | Heart team decision tree.** Decision criteria of the heart team. All decisions remain individualized decisions after careful discussion of comorbidities and anatomical factors. SAVR = surgical aortic valve replacement; TAVR = transcatheter aortic valve replacement.

## Table 1 | Baseline characteristics of the cohort

| Parameter | All (N = 16) | TAVR (N = 9) | SAVR (N = 7) |
|---|---|---|---|
| Age (years) | 57 (44; 60) | 56 (44; 59) | 57 (46; 66) |
| Sex | | | |
| male | 11 (69%) | 5 (56%) | 6 (86%) |
| female | 5 (31%) | 4 (44%) | 1 (14%) |
| Weight (kg) | 69.5 (59.8; 78.1) | 72 (66.3; 78) | 69 (57.5; 82.5) |
| Height (cm) | 174 (165; 179) | 168 (163; 179) | 175 (173; 178) |
| BMI (kg/m²) | 23.7 (21.7; 27.1) | 23.7 (23.3; 26.5) | 22.5 (20; 29) |
| Chronic dialysis prior to procedure | | | |
| Yes | 6 (38%) | 2 (22%) | 4 (57%) |
| No | 10 (62%) | 7 (78%) | 3 (43%) |
| Insulin-dependent diabetes mellitus | | | |
| Yes | 4 (25%) | 3 (33%) | 1 (14%) |
| No | 12 (75%) | 6 (67%) | 6 (86%) |
| Pre-procedural medication: | | | |
| ASS | 6 (38%) | 1 (11%) | 5 (71%) |
| Clopidogrel | 1 (6%) | 1 (11%) | 0 (0%) |
| Vitamin K antagonist | 4 (25%) | 4 (44%) | 0 (0%) |
| direct oral anticoagulation | 1 (6%) | 1 (11%) | 0 (0%) |
| Statin | 11 (69%) | 6 (67%) | 5 (71%) |
| Coronary angiogram prior to procedure | | | |
| No stenosis in need of an intervention | 14 (88%) | 7 (78%) | 6 (86%) |
| Implantation of drug-eluting stent | 1 (6%) | 1 (1%) | 0 (0%) |
| Simultaneous intervention planned with TAVR/SAVR | 0 (0%) | 0 (0%) | 0 (0%) |
| Secondary coronary intervention planned after TAVR/SAVR | 1 (6%) | 0 (0%) | 1 (14%) |

Categorical data is displayed as numbers (%), continuous data is stated as median (interquartile range).
*HTX* heart transplantation, *DCMP* dilated cardiomyopathy, *CAD* coronary artery disease, *TAVR* transcatheter aortic valve replacement, *SAVR* surgical aortic valve replacement.

## Reporting summary

Further information on research design is available in the Nature Portfolio Reporting Summary linked to this article.

## Results

Of 2054 patients who underwent orthotopic heart transplantation between 1986 and 2023 at the Deutsches Herzzentrum der Charité, Berlin, Germany, 39 patients (1.9%) developed moderate or severe aortic valve dysfunction. We report on 16 patients who underwent surgical aortic valve replacement (SAVR, N = 7) or transcatheter aortic valve replacement (TAVR, N = 9) after heart transplantation (Fig. 1) between 1998 and 2024. Median time between heart transplantation and the aortic valve procedure was 13.4 years (IQR 5.4; 17.5 years) (TAVR 17 years (IQR 10; 18 years), respectively, SAVR 8.5 years (IQR 3; 13.5 years)). Median time from first diagnosis of at least moderate aortic valvular dysfunction to TAVR was 26 months (IQR 5.5; 47 months). Isolated aortic valve regurgitation occurred in 5 (31%) patients, of which in 3 it was caused by infectious endocarditis. Eleven patients (69%) suffered from aortic valve stenosis with (N = 7 (44%)) or without (N = 4 (25%)) concomitant regurgitation. General baseline characteristics are displayed in Table 1, baseline characteristics specific to the aortic valve in Table 2 and laboratory results in Table 3. Pre-procedural heart failure was present in 38% of the patients and 75% of the patients were in a NYHA functional class III or IV. Use of vasoactive therapy was documented in none of the TAVR patients and 2 (29%) of the SAVR patients, both of whom were treated with norepinephrine.

### Technical considerations

Nine (56%) patients underwent TAVR, seven (78%) with a transfemoral access (TF-TAVR), and in two (22%) patients a transapical access site was used (TA-TAVR). The first TA-TAVR was done in 2014, the first TF-TAVR in 2015. The use of a transapical access was based on the absence of a dedicated TF-TAVR device for the treatment of isolated aortic valve regurgitation at the time in one patient (patient T8) and the presence of severe peripheral artery disease precluding a safe retrograde access in the other patient (patient T3).

Access site for surgical aortic valve replacement was a full median sternotomy in all cases (N = 7 (100%)). Three (43%) patients received a biological valve prosthesis, in 4 (57%) cases a mechanical valve replacement was chosen. One (14%) patient underwent multi-valve surgery with concomitant biological mitral and tricuspid valve replacement. Median procedural duration was 5 h (IQR 3.9 h; 5.13 h) and median cardiopulmonary bypass time was 129 min (IQR 108 min; 174 min). Due to the complexity of the surgery, the patient with triple valve replacement was cannulated peripherally prior to sternotomy, in all other patients a central cannulation was achieved. Two patients underwent normothermic perfusion (36 °C), 2 patients underwent moderate hypothermia with 32 °C and three patients with ≤30 °C. Supplementary Data 2 displays specifics on every individual case.

All patients underwent coronary angiograms and myocardial biopsy prior to intervention/surgery to exclude cardiac deterioration due to rejection or relevant cardiac allograft vasculopathy as a differential diagnosis.

**Table 2 | Baseline characteristics of the aortic valve condition**

| Parameter | All (N = 16) | TAVR (N = 9) | SAVR (N = 7) |
|---|---|---|---|
| NYHA class prior to valve surgery | | | |
| I | 0 (0%) | 0 (0%) | 0 (0%) |
| II | 4 (25%) | 1 (11%) | 3 (43%) |
| III | 8 (50%) | 7 (78%) | 1 (14%) |
| IV | 4 (25%) | 1 (11%) | 3 (43%) |
| Aortic valve pathology | | | |
| Severe stenosis | 10 (63%) | 7 (77%) | 3 (43%) |
| Moderate stenosis | 1 (6%) | 1 (11%) | 0 (0%) |
| Aortic valve regurgitation 0° | 4 (25%) | 2 (22%) | 2 (29%) |
| Aortic valve regurgitation I° | 4 (25%) | 3 (33%) | 1 (14%) |
| Aortic valve regurgitation II° | 3 (19%) | 2 (22%) | 1 (14%) |
| Aortic valve regurgitation III° | 3 (19%) | 2 (22%) | 3 (43%) |
| Infective endocarditis | 3 (19%) | 0 (0%) | 3 (43%) |
| Combined dysfunction | 7 (44%) | 6 (67%) | 1 (14%) |
| Isolated aortic valve stenosis | 4 (25%) | 2 (22%) | 2 (29%) |
| Isolated aortic valve regurgitation (incl. IE) | 5 (31%) | 1 (11%) | 4 (57%) |
| Preoperative condition | | | |
| no signs of cardiac decompensation | 9 (56%) | 6 (67%) | 3 (43%) |
| signs of decompensation* | 6 (38%) | 3 (33%) | 3 (43%) |
| Information not available | 1 (6%) | 0 (0%) | 1 (14%) |
| need of vasoactive therapy | 2 (12.5%) | 0 (0%) | 2 (29%) |
| LVEF (%) (preoperative) | 57 (51; 60) | 55 (41; 60) | 60 (57; 60) |
| LVEDD (mm) | 52 (43; 54) | 52 (39; 53) | 53 (48; 58) |
| E/A | | | |
| > 2 | 4 (25%) | 3 (33%) | 1 (14%) |
| <2 | 2 (12.5%) | 2 (22%) | 0 (0%) |
| NA | 10 (62.5%) | 4 (45%) | 6 (86%) |
| Euroscore II | 7.2 (4.6; 13.2) | 7.7 (6.1; 10.6) | 5.0 (4.6; 13.3) |

Categorical data is displayed as numbers (%), continuous data is stated as median (interquartile range).

*HTX* heart transplantation, *LVEF* left ventricular ejection fraction, *LVEDD* left ventricular end-diastolic diameter, *E/A* E wave velocity/A wave velocity of the mitral valve, *CRP* C-reactive protein, *TAVR* transcatheter aortic valve replacement, *SAVR* surgical aortic valve replacement.

* signs of decompensation: documentation of pre-procedural pleural effusion, ascites, progressive peripheral edema, worsening of dyspnea to NYHA III or IV, pulmonary edema with or without consecutive pneumonia or new-onset dialysis due to acute (worsening of) cardio-renal syndrome.

## Survival

30-day survival was 100%. In-hospital mortality was $N = 2$ (22%) due to infection; both patients underwent TAVR as a rescue therapy in cardiogenic shock. None of the patients developed an access site infection; one patient suffered from spontaneous bacterial peritonitis in combination with pneumonia, and one patient had a pneumonic focus. Further specifics are displayed in Table 4: Survival of patients with aortic valve replacement and Fig. 3: Survival of patients with aortic valve replacement.

## Peri-procedural details

Both SAVR and TAVR procedures were conducted with technical and device success in all cases according to VARC-3 criteria (Table 5: Post-procedural data). Eight (50%) patients had an uneventful postoperative course. The VARC-3 composite safety endpoint was achieved in seven TAVR patients and 4 SAVR patients. One TAVR patient suffered from sub-acute ischemic stroke on day 3, and one TAVR patient underwent post-interventional pacemaker implantation due to grade III° AV block on the fifth postoperative day. In the SAVR group, two patients needed

**Table 3 | Laboratory results**

| Parameter | All (N = 16) | TAVR (N = 9) | SAVR (N = 7) |
|---|---|---|---|
| **Pre-procedural laboratory results** | | | |
| Creatinine mg/dl | 2.6 (1.95; 4.55) | 2.1 (1.9; 4) | 3.3 (2.6; 4.7) |
| Hemoglobin g/dl | 11.8 (9.3; 12.7) | 10.7 (9.5; 12.3) | 11.9 (8.8; 12.8) |
| CRP mg/dl | 0.95 (0.3; 1.85) | 0.9 (0.3; 2.6) | 1 (0.55; 1.3) |
| White blood cell count (K/μl) | 6.1 (4.75; 8.1) | 6.8 (4.4; 8) | 5.95 (5.35; 7.68) |
| Sodium (mmol/l) | 138 (134.5; 140) | 137 (133; 139) | 142 (140; 143) |
| Bilirubin (mg/dl) | 0.62 (0.43; 0.95) | 0.62 (0.43; 0.92 | 0.87 (0.53; 1.27) |
| **Post-procedural laboratory results** | | | |
| Creatinine (mg/dl) New-onset dialysis | 2.18 (1.75; 3.95) 3 (19%) | 2.10 (1.60; 3.10) 2 (22%) | 2.74 (2.02; 4.35) 1 (14%) |
| Hemoglobin (g/dl) | 8.7 (8.2; 9.9) | 8.5 (8.1; 8.7) | 9.7 (9; 10) |
| CRP (mg/dl) | 4.9 (3.1; 7.6) | 5.8 (4.9; 8.6) | 2.5 (1.8; 3.3) |
| White blood cell count (k/μl) | 8 (4.9; 12.5) | 7.4 (4.9; 9.5) | 11.2 (8.5; 12.8) |
| Sodium (mmol/l) | 139 (136; 141) | 140 (138; 141) | 138 (135; 140) |
| Bilirubin (mg/dl) | 0.66 (0.50; 0.83) | 0.55 (0.50; 0.81) | 0.75 (0.59; 1.1) |

Categorical data is displayed as numbers (%), continuous data is stated as median (interquartile range).
*CRP* C-reactive protein.

**Table 4 | Survival of patients with aortic valve replacement**

| Parameter | All (N = 16) | TAVR (N = 9) | SAVR (N = 7) |
|---|---|---|---|
| Survival time after valve surgery (years) | 1.5 (0.9; 2.6) | 1.2 (0.3; 2.4) | 1.5 (1.3; 2.7) |
| Survival time after HTX (years) | 17 (11; 19) | 18 (16; 18) | 12 (8; 19) |
| All-cause Mortality | | | |
| Peri-procedural (≤ 30 days) | 0 (0%) | 0 (0%) | 0 (0%) |
| Early (> 30 days and ≤ 1 year) | 3 (19%) | 3 (33%) | 0 (0%) |
| Late (> 1 year and ≤ 3 years) | 6 (60%) | 2 (50%) | 4 (67%) |
| Censored ≤ 90 days | 1 (6%) | 0 (0%) | 1 (14%) |
| Censored > 1 year and <3 years | 3 (19%) | 2 (22%) | 1 (14%) |
| Cardiovascular mortality (at 1 year) | 0 (0%) | 0 (0%) | 0 (0%) |
| Cardiovascular mortality (> 1 year) | | | |
| acute graft failure | 2 (13%) | 0 (0%) | 2 (33%) |
| valve-related mortality | 0 (0%) | 0 (0%) | 0 (0%) |
| Cause of death | | | |
| Graft failure/rejection | 3 (27%) | 0 (0%) | 3 (50%) |
| Sepsis/infection | 5 (46%) | 4 (80%) | 1 (17%) |
| Unknown | 3 (27%) | 1 (20%) | 2 (33%) |

Categorical data is displayed as numbers (%), continuous data is stated as median (interquartile range).
*HTX* heart transplantation, *TAVR* transcatheter aortic valve replacement, *SAVR* surgical aortic valve replacement, *NAR* number at risk.

postoperative mechanical circulatory support with an intra-aortic balloon pump (IABP), in both patients, weaning was successful. One patient had to be readmitted for re-thoracotomy due to pericardial hematoma. One patient suffered a postoperative stroke while on IABP support. Neither early- nor late-onset infectious complications of the sternal wound occurred in the SAVR cohort. Post-interventional pneumonia developed in 3 (33%) TAVR

**Fig. 3 | Survival after aortic valve replacement.** Three-year survival after aortic valve replacement. Number of patients at risk is stated in number (percent) and the cumulative number of events is state as a number.

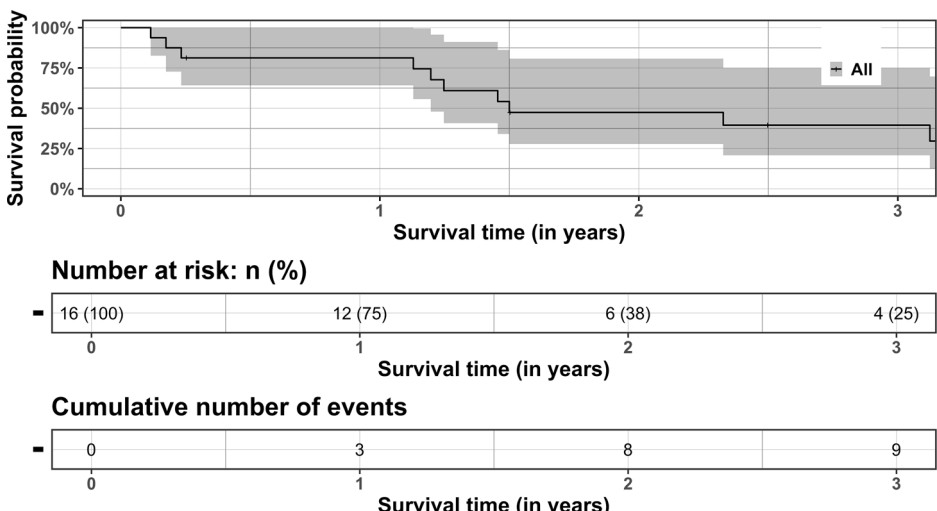

## Table 5 | Post-procedural data

| Parameter | All (N = 16) | TAVR (N = 9) | SAVR (N = 7) |
|---|---|---|---|
| Paravalvular leakage | | | |
| None | 15 (94%) | 8 (89%) | 7 (100%) |
| Mild | 1 (6%) | 1 (11%) | 0 (0%) |
| Severe | 0 (0%) | 0 (0%) | 0 (0%) |
| Mean pressure gradient (mmHg) | 11 (9; 18) | 11 (7; 16) | 14.5 (11; 19) |
| Diameter of valve prothesis (mm) | 25 (23; 26) | 26 (25; 29) | 23 (23; 24) |
| LVEF (%) | 58 (55; 60) | 55 (45; 60) | 60 (58; 61) |
| Length of hospital stay (days) | 29 (17; 63) | 31 (10; 61) | 27 (21; 67) |
| Transfusion of packed red blood cells | 2 (0; 4) | 1 (0; 2) | 5 (3; 6) |
| NYHA class after valve surgery / intervention | | | |
| I | 11 (69%) | 5 (56%) | 6 (86%) |
| II | 1 (6%) | 1 (11%) | 0 (0%) |
| III | 1 (6%) | 0 (0%) | 1 (14%) |
| IV | 0 (0%) | 0 (0%) | 0 (0%) |
| NA | 3 (19%) | 3 (33%) | 0 (0%) |
| Subjective improvement | 10 (62%) | 5 (55%) | 5 (71%) |
| Postoperative course | | | |
| Uneventful | 8 (50%) | 4 (44%) | 4 (57%) |
| Stroke | 2 (13%) | 1 (11%) | 1 (14%) |
| Pacemaker implantation | 1 (6%) | 1 (11%) | 0 (0%) |
| Access site complication | 1 (6%) | 1 (11%) | 0 (0%) |
| MCS | 2 (13%) | 0 (0%) | 2 (29%) |
| Postoperative infection | 4 (25%) | 3 (33%) | 1 (14%) |

Categorical data is displayed as numbers (%), continuous data is stated as median (interquartile range).
*LVEF* left ventricular ejection fraction, *CRP* C-reactive protein, *NYHA* New York Heart Association, *NA* not applicable, *TAVR* transcatheter aortic valve replacement, *SAVR* surgical aortic valve replacement, *MCS* mechanical circulatory support.

patients and 1 SAVR patient. The periprocedural management of the immunosuppressive therapy is described in the Supplementary Fig. S1: Immunosuppressive management.

The patients' functional status improved in 11 (69%) cases to NYHA class I, 1 (6%) TAVR patient to NYHA class II and 1 (6%) SAVR patient improved from NYHA Class IV to NYHA Class III. In 3 (19%) patients no follow-up data regarding the NYHA Class was available (Tables 2, 5 and Supplementary Data 2: Peri-procedural details). Pre- and post-procedural laboratory results are displayed in Table 3.

## Discussion

Aortic valve dysfunction is a rare but relevant long-term complication after heart transplantation. Feasible treatment options include surgical and interventional aortic valve replacement, the latter being increasingly preferred due to the fast-growing evidence and experience in non-transplant patients[5,6]. However, SAVR is a viable option in cases of endocarditis, hostile landing zone and/or cases in need of concomitant cardiac surgical procedures.

With an estimated prevalence of 1% and a median age of 57 years in our cohort, aortic stenosis appears to occur at an accelerated rate in heart transplanted patients compared to the literature (0.2% in patients between 50 and 60 years old, 1.3% in patients between 60 and 70 years old)[10]. Inflammatory processes on the endothelium in transplanted patients may contribute to the earlier development of aortic valve stenosis, since the etiology of calcified aortic valve degeneration is thought to be caused by long-term endothelial stress[10]. As in the literature, the leading cause for isolated aortic valve regurgitation was infective endocarditis[3,11]. As a specimen, we proved methicillin-resistant Staphylococcus aureus in one case and Enterococcus faecalis in another case; one case remained culture-negative. The literature reported 14% culture-negative endocarditis, Staphylococcus aureus in 42% of the cases (13% methicillin resistant) and E. faecalis in 19.4% of the cases, with bacterial endocarditis in a review concerning infectious endocarditis after heart transplantation[12].

The choice of the treatment remains an individualized team decision and is based on the principles of anatomical, pathophysiological, and device-related considerations applicable to SAVR and TAVR[13]. Timely aortic valve replacement before the onset of heart failure seems to be critical in transplant patients, as an impaired clinical state may severely impact outcomes[14]. A considerable number of patients (38%) in this cohort showed signs of severe cardiac deterioration prior to the procedure.

In general, procedural planning, device selection, and implant technique for TAVR in heart transplant patients align with current standards for the non-transplant population. A transfemoral access was the preferred approach for TAVR. If the transfemoral approach was not feasible, percutaneous transaxillary access, to avoid impaired wound healing or infectious complications was chosen[4,15,16]. Devices were selected based on annular size and root anatomy, with no specific anatomical challenges arising from the post-transplant situation, particularly not the aortic anastomosis (angle/kinking). Similar to the non-transplant population, a TAVR prosthesis that ensures optimal transvalvular gradients, paravalvular sealing, and low

pacemaker risk was chosen to maintain myocardial function. For younger patients in whom TAVR may be intended as a bridge to future cardiac re-transplantation as a lifetime management concept, the selection of a transcatheter heart valve with a short stent frame may be preferred to preserve aortic wall tissue for subsequent aortic anastomosis after explantation. Coronary access and neocommissural alignment/coronary alignment are particularly important in patients with allograft vasculopathy and repeated coronary interventions. Radial access is used for aortic root pigtail, and pacing is tailored to valve type: transvenous right ventricular pacing for balloon-expandable valves and TAVR-wire left ventricular pacing for self-expanding valves, when this seems sensible. A transcatheter approach may be beneficial in a population with chronic immunosuppressive therapy, as wound healing problems and post-procedural infections can be avoided. Despite the minimally invasive access of TAVR, adjustment of mycophenolate-mofetil/everolimus-based immunosuppression should be discussed to reduce the risk of peri-procedural infections. Pneumonia can be triggered by heart failure with pulmonary congestion and seems to contribute to early mortality after TAVR in this cohort. Transfusion thresholds and pre-operative correction of anemia need to be discussed as part of patient-blood management in general and in the context of potential sensitization in particular[17]. As repeated coronary angiography is routinely performed after heart transplantation to monitor cardiac allograft vasculopathy, femoral access for TAVR should be ultrasound-guided to minimize the risk of access site complications. In this context, it is also important to highlight the need to preserve safe coronary access when performing TAVR to enable future coronary angiography to diagnose and treat transplant vasculopathy.

Redo-surgery in transplanted patients is a high-risk procedure, which is not adequately represented by classic surgical risk scores, such as the STS score or Euroscore II, since heart transplanted patients were excluded from validation of these scores[18,19]. Comorbidities, re-thoracotomy and the immunosuppressive therapy increase the complexity of the procedure and the risk of infection. The high one-year survival of patients with SAVR after heart transplantation in this cohort is indicative of the rigorous patient selection.

Careful discussion with the patient about the long-term prognosis after SAVR and TAVR is necessary, especially since the literature does not provide any reliable data concerning the long-term outcomes after SAVR or TAVR in the setting of HTX, and a comparison to conservative management is lacking. Based on our limited data, an improvement of functional status (NYHA class I or II) can be expected in 75% of the patients.

It is to be acknowledged that the current study has a number of limitations. First, by its nature, it is subject to the restrictions of a retrospective study. Despite reporting all cases from our center over a period of nearly 40 years, the number of patients is not sufficient for further statistical analysis or group comparison. Data from a multi-center registry would be required to obtain a larger and more complete case overview. Advances in medical techniques additionally bias our analysis, since SAVR was mostly performed between 1998 and 2010, and TAVR was only performed after 2014. Despite an early electronic health record system, documentation of the treatment of the SAVR patients is incomplete, especially regarding the medication used during the hospital stay (i.e., vasoactive therapy prior to surgery, transfusions).

Data on disease dynamics and on conservative treatment as a comparison remain scarce. The three-year survival rate is low compared to the PARTNER trial, highlighting that these are the highest risk patients[7]. Data on conservative treatment or even a prospective comparison will not be available since leaving the aortic valve untreated is regarded as unethical, making such descriptive analysis even more important. Second, we would like to highlight that these patients with a median survival of 17 years (IQR 11;19 years) after heart transplantation already outlived their median life expectancy after heart transplantation at the time of the procedure, which may influence the low post-procedural survival rate. Therefore, it is not possible to conclude about treatment futility regarding survival based on this data.

To the best of our knowledge, this study presents the largest single-center experience with surgical and interventional aortic valve procedures after orthotopic heart transplantation reported so far. The scope of future research may include data on disease dynamics and comparison to conservative treatment, possibly in the form of a multi-center registry. Pathological examination of resected valves and structured echocardiographic monitoring may help to conclude about differences in the pathophysiology of aortic valve degeneration in heart transplantation compared to non-transplant patients.

## Conclusion
Aortic valve dysfunction is a rare but relevant long-term complication after heart transplantation. TAVR is the preferred treatment in our center, but SAVR is feasible for individuals ineligible for an interventional approach. The peri-procedural risks are higher than in non-transplant patients; however, 75% of the patients reported a subjective improvement in their well-being and functional status after the procedure.

## Data availability
The complete data cannot be shared openly due to privacy concerns from the small number of participants. For further request, please contact the corresponding author at luise.roehrich@dhzc-charite.de. The source data for Figs. 1 and 3 can be found in Supplementary Data 4 "Source data of the figures". There is no source data applicable for Fig. 2 due to the nature of a flowchart. The source data of figure S1 is displayed within the figure.

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

## Author contributions

Corresponding author: L.R. Study design, data collection, draft of the manuscript: L.R. and H.H. Supervision of the project: A.U., F.S. Compilation of the manuscript: L.R., H.H., C.K., F.H., C.K., M.K., N.M., L.K.F., I.A.J., S.O., H.D., V.F., A.U., and F.S.

## Funding

## Competing interests

The following authors declare no competing interests: L.R., H.H.i., C.K., F.H., C.K.I., M.K., N.M., L.K.F. The authors declare the following competing interests outside of the submitted work: I.A.J. Speakers honoraria from astrazeneca, Abbott, Abiomed. S.O. received institutional research and study funds from Novartis Pharma GmbH and institutional research, study and educational grants, speaker fees and advisory board fees from Abiomed. H.D. Speaker's fees, advisory board: Abbott, Edwards; research support: Abbott. V.F. institutional financial activities in relation to the following: educational grants (including travel support), fees for lectures and speeches, fees for professional consultation, and research and study funds with the following commercial entities: Medtronic GmbH, Biotronik S.E. & Co, Abiomed GmbH, Abbott GmbH & Co, KG, Boston Scientific, Edwards Lifesciences, Berlin Heart, Novartis Pharma GmbH, JOTEC/CryoLife GmbH, LivaNova, and Zurich Heart. A.U. Physician proctor to Edwards Lifesciences, Medtronic and JenaValve. F.S. received institutional grants from Novartis, Abbott, non-financial support from Medtronic and institutional fees (speaker honoraria) from Abbott, Bayer, Novartis and Abiomed outside of the submitted work
