## [Transparent Peer Review file · Communications Medicine]

Surgical and transcatheter aortic valve replacement after orthotopic heart transplantation

Corresponding Author: Dr Luise Roehrich

Version 0:

Reviewer comments:

Reviewer #1

(Remarks to the Author)

Thank you very much for giving me this opportunity to review this manuscript.

The authors demonstrated the outcomes of aortic valve surgery following heart transplantation.

As they pointed out, the aortic valve disease after heart transplantation is relatively rare, and it would be important to know the clinical outcomes of these cases. I have some questions and suggestions for the authors.

1. line 9-10 Infective endocarditis should be removed. This might be typo.
2. More detail information regarding preoperative echocardiography would be helpful to understand how the aortic valves are deteriorated. For example, the mean gradient of aortic valve, peak velocity of aortic valve, etc.
3. In terms of cause of death, most of them in TAVR group were infection/sepsis, while those in SAVR group were graft failure. Do the authors have any explanation why the cause of death are different in two groups? Are there any specific reasons why the graft failure rate was high in SAVR group?
4. In Table3, the paravalvular regurgitation of SAVR was 100% in all of them. I believe this is typo.
5. I think the pathological findings of these aortic valves might be very informative because the aortic valve disease after heart transplantation is relatively rare. Are the authors able to show the pathological findings of the resected aortic valve? Is there any differences from common aortic valve stenosis and regurgitation?
6. Could the authors demonstrate the time course of progression of aortic valve stenosis and aortic valve regurgitation? It might be helpful for the physicians to know how quick these diseases progress.

Reviewer #2

(Remarks to the Author)

Thank you for submitting such a detailed manuscript. It effectively reflects your experience with aortic valve intervention following orthotopic heart transplantation. As expected, the sample size is small, making this a valuable case series for the cardiac community.

I have a few specific points of interest:

1. I would be interested in details regarding cases with AV dysfunction cases that did not proceed to intervention.
2. As a TAVI operator, I would appreciate more insight into the technical aspects of the procedure, including prosthesis selection, key technical considerations, and procedural challenges.

Minor comments:

- There is inconsistent use of commas within numerical values (lines 20–35).
- The conclusion is too lengthy and could be more concise.

Reviewer #3

(Remarks to the Author)

I thank the authors for an insightful study that will benefit the readers, due to the paucity of evidence surrounding valvular dysfunction and treatment following heart transplantation (HTX). I endorse the future publication of this paper though I do have a few suggestions/queries that may strengthen the article.

Infective endocarditis appears to be a common cause of aortic regurgitation post HTX. Were there any specific pathogens associated with this?

Were there any healing issues or infective complications following sternotomy with concurrent anti-rejection medications?

What was the anti-rejection regimen peri-operatively? The authors have included a table with the medications, but knowledge regarding commencement timing of medication and order of commencement would be useful for the readers. An update to the table should suffice.

Three-year survival rate following aortic valve replacement was low at 25%. In fact, the two-year survival rate was lower to the non-TAVI arm of the PARTNER IB trial. This highlights an important question that needs further answers: is treatment futile?

In patients requiring aortic valve replacement, what was the rate of severe coronary stenoses? How many of these patients required revascularisation?

Version 1:

Reviewer comments:

Reviewer #1

(Remarks to the Author)

The authors addressed the points I raised in reviewer comment. I do not have any further comments.

Reviewer #2

(Remarks to the Author)

This is a well-written manuscript that serves as a valuable contribution—a relatively “large” case series on a rare and underreported topic. It offers important and informative insights for the cardiology community.

That said, the conclusion section remains too wordy and would benefit from more concise and focused messaging to enhance clarity and impact.

Reviewer #3

(Remarks to the Author)

Thank you for addressing my queries. I do not have any further issues with this manuscript.

Dear Reviewers, dear editorial team,

Thank you for the time and effort to provide us with such an insightful review to our manuscript. We feel strongly that the manuscript benefits from clarification of the risen points:

Reviewer #1 (Remarks to the Author):

Thank you very much for giving me this opportunity to review this manuscript.

The authors demonstrated the outcomes of aortic valve surgery following heart transplantation. As they pointed out, the aortic valve disease after heart transplantation is relatively rare, and it would be important to know the clinical outcomes of these cases. I have some questions and suggestions for the authors.

1. line 9-10 Infective endocarditis should be removed. This might be typo.

Thank you for the suggestion. Since infective endocarditis is a special cause for aortic regurgitation and the treatment is limited to SAVR, we thought this information would be helpful for the reader. However, we appreciate this point of view and removed the information from this sentence.

2. More detail information regarding preoperative echocardiography would be helpful to understand how the aortic valves are deteriorated. For example, the mean gradient of aortic valve, peak velocity of aortic valve, etc.

Thank you for this great suggestion. The cohort itself and the echocardiographic data available is very heterogenic, since the inclusion time is spread over such a long time and medicine progressed immensely since 1986. We included a new case-to-case table with the available echocardiographic data in the supplementary material (table S3), since a descriptive analysis for these parameters is just not feasible in such a small, heterogenic cohort. Part of the SAVR patients underwent surgery prior to 2000, therefore there are no pictures in our electronic system and we have to rely on the reports only. Post-imaging processing is therefore not possible in all cases. Furthermore, imaging techniques advanced immensely since then. We hope for more complete data in the future, as discussed in the new section "scope of future research".

3. In terms of cause of death, most of them in TAVR group were infection/sepsis, while those in SAVR group were graft failure. Do the authors have any explanation why the

cause of death are different in two groups? Are there any specific reasons why the graft failure rate was high in SAVR group?

A possible bias may be the changes in immunosuppressive therapy regimes over time: As displayed in figure S1, TAVR patients received Tacrolimus and Everolimus containing regimes more often at time of the procedure, while most of the SAVR patients stayed on Ciclosporin and an antimetabolite. This may confound both, infectious complications and graft failure. However, due to the low case rate, an appropriate analysis is not possible and this stays a pure hypothesis at this time. Therefore, we refrained from commenting on this in the manuscript at this time and hope for further data in the future.

4. In Table 3, the paravalvular regurgitation of SAVR was 100% in all of them. I believe this is typo.

Thank you for this correction. We changed the numbers and percent as appropriately.

5. I think the pathological findings of these aortic valves might be very informative because the aortic valve disease after heart transplantation is relatively rare. Are the authors able to show the pathological findings of the resected aortic valve? Is there any differences from common aortic valve stenosis and regurgitation?

We completely agree, that a pathological examination of the valves would be very interesting. However, since this is not standard of care in our hospital and TAVR increasingly replaces SAVR, unfortunately this information is not available. Lu et al found in their post-mortem study that 2 out of 64 HTX patients had a valvular heart disease, however further pathological findings of the valves are not described in their manuscript since the study scope was cardiac allograft vasculopathy. However, we will include a pathological examination for SAVR patients in the future. As far as we know, there are no pathological differences except it seem to occur accelerated possibly due to immunological effects on the endocardium (compare section discussion). We included this point in the new section "scope of future research".

6. Could the authors demonstrate the time course of progression of aortic valve stenosis and aortic valve regurgitation? It might be helpful for the physicians to know how quick these diseases progress.

Thank you for this suggestion, this is an important, yet difficult to assess point. The median time since first diagnosis of an at least moderate valve dysfunction for TAVR patients is stated in the

results (first paragraph, description of the cohort). Unfortunately, due to the lack of regular echocardiography documentation in historical patients (it seems echocardiography was only done, if the patients are already symptomatic at this time), for SAVR patients this information is mostly not available. We hope to provide more data on this in the future, as regular echocardiography as monitoring is now standard of care in our center and widely available. We included this point in the study limitations and new section “scope of future research”.

Reviewer #2 (Remarks to the Author):

Thank you for submitting such a detailed manuscript. It effectively reflects your experience with aortic valve intervention following orthotopic heart transplantation. As expected, the sample size is small, making this a valuable case series for the cardiac community.

I have a few specific points of interest:

1. I would be interested in details regarding cases with AV dysfunction cases that did not proceed to intervention.

Thank you for this comment, this is a fascinating question. Unfortunately, it is not possible to provide data to this question at the moment. To our knowledge all patients with a documented indication for treatment also underwent treatment, especially since TAVR has become a good option for heart transplanted patients in our center. However, we plan to start a registry concerning valvular dysfunction in heart transplanted patients in our center, hopefully we can provide more data to this question in the future. This would also help to determine disease dynamics, as suggested by reviewer 1. We included these points in the new section “scope of future research”.

2. As a TAVI operator, I would appreciate more insight into the technical aspects of the procedure, including prosthesis selection, key technical considerations, and procedural challenges.

Thank you for your comment. We included the following details in the discussion in the section about the periprocedural considerations (lines 176 ff):

Generally, procedural planning, device selection, and implant technique for TAVR in heart transplant patients align with current gold standards for the non-transplanted population. We

select devices based on annular size and root anatomy, with no specific anatomical challenges arising from the post-transplant situation, particularly not the aortic anastomosis (angle/kinking). Transfemoral access is the preferred approach, and when not feasible, we use transaxillary access percutaneously to avoid wound healing or infectious complications. Similar to the non-transplant population, we prioritize selecting a TAVR prosthesis that ensures optimal transvalvular gradients, paravalvular sealing, and low pacemaker risk, as these factors are crucial for myocardial and allograft function and especially the pacemaker risk may impede survival. For younger patients in whom TAVR may be intended as bridge to future cardiac re-transplantation as lifetime management concept, the selection of a transcatheter heart valve with a short stent frame may be preferred to preserve aortic wall tissue for subsequent aortic anastomosis after explant. Coronary access and neocommissural alignment/coronary alignment are particularly important in patients with allograft vasculopathy and repeated coronary interventions. For anesthesia, we use awake sedation with remifentanyl and local anesthesia. Radial access is used for aortic root pigtail, and pacing is tailored to valve type: transvenous right ventricular-pacing for balloon-expandable valves and TAVR-wire left ventricular-pacing for self-expanding valves, when this seems sensible.

Minor comments:

- **There is inconsistent use of commas within numerical values (lines 20–35).**
- **The conclusion is too lengthy and could be more concise.**

Thank you for both comments. Interpunctuation was changes appropriately and we shortened the conclusions

Reviewer #3 (Remarks to the Author):

I thank the authors for an insightful study that will benefit the readers, due to the paucity of evidence surrounding valvular dysfunction and treatment following heart transplantation (HTX). I endorse the future publication of this paper though I do have a few suggestions/queries that may strengthen the article.

Infective endocarditis appears to be a common cause of aortic regurgitation post HTX. Were there any specific pathogens associated with this?

Thank you for the question! Of the three endocarditis cases, we had evidence for case with methicillin-resistant *Staphylococcus aureus* and one case with an *Enterococcus faecalis*, one case stayed culture-negative (information is included in Table S2 of the supplementary material). Jordan et al supplied us with a very conclusive review about endocarditis in patients after heart transplantation: 14% stayed culture-negative, they found *Staph aureus* in 42% of the cases (13% methicillin resistant) and *E. faecalis* in 19.4% of the cases with bacterial endocarditis and named fungi as an additional organism of endocarditis special to immunosuppressed patients, latter was associated with an increased mortality. We included this information in the discussion (lines 163 ff).

Were there any healing issues or infective complications following sternotomy with concurrent anti-rejection medications?

Thank you for this suggestion! Despite immunosuppressive therapy, our patients did not experience early- or late-onset infectious complications of the sternotomy. We included this detail in the section “results – peri-procedural details” (line 143)

What was the anti-rejection regimen peri-operatively? The authors have included a table with the medications, but knowledge regarding commencement timing of medication and order of commencement would be useful for the readers. An update to the table should suffice.

Thank you for this proposition. We extended Figure S1 with the available information.

Three-year survival rate following aortic valve replacement was low at 25%. In fact, the two-year survival rate was lower to the non-TAVI arm of the PARTNER IB trial. This highlights an important question that needs further answers: is treatment futile?

This is a very good question, which unfortunately is not possible to be conclusively answered based on a case series. In our opinion, several points are important in this discussion and should be focus of future research:

Data on disease dynamics and on conservative treatment as a comparison is scarce at the moment, but will be scope of future research at our center. The three-year survival rate is low, highlighting that these are high-risk patients. However, to conclude about futility, we need to a)

exclude the low survival rate to be an effect of the low case number and b) we need data on conservative treatment as a comparison, which is not available at the time. Second, we would like to highlight, that these patients with a median survival of 17 years (IQR 11;19 years) after heart transplantation already outlived their median life expectancy after heart transplantation, which may influence the low post-procedural survival rate. We amended this in the study limitations and in the new section “scope of future research”.

Additionally, as discussed in the manuscript, timing of the procedure is important concerning the futility of treatment discussion: This procedure (TAVR and SAVR) qualifies as high-risk in patients after heart transplantation. 75% of the patients were highly symptomatic (NYHA class III or IV) at time of intervention/surgery. Therefore, patients may benefit from an earlier intervention especially with TAVR regarding the postoperative/-interventional survival (if we may refer to the results of the EARLY TAVR trial Genereux et al).

In patients requiring aortic valve replacement, what was the rate of severe coronary stenoses? How many of these patients required revascularisation?

Thank you for this important question, especially since concomitant need of coronary revascularization would possible change indication towards SAVR. However, none of the patients required simultaneously a revascularization during surgery/intervention. 26 % of the patients underwent intervention for cardiac allograft vasculopathy in the treatment course prior to the aortic valve dysfunction (please compare Table S1 “cardiac allograft vasculopathy”). Every patient underwent coronary angiogram prior to procedure, one patient required an implantation of a drug-eluting stent in the left main coronary artery. One patient had peripheral stenosis and a secondary coronary intervention after SAVR was planned. We included this information in Table 1: Baseline characteristics.

Yours sincerely,

Luise Roehrich

Dear Reviewers, dear Editorial Team,

Thank you for your time and effort to improve our manuscript further.

We addressed the last risen feedback point from Reviewer 2:

Reviewer #2 (Remarks to the Author):

This is a well-written manuscript that serves as a valuable contribution—a relatively “large” case series on a rare and underreported topic. It offers important and informative insights for the cardiology community.

That said, the conclusion section remains too wordy and would benefit from more concise and focused messaging to enhance clarity and impact.

Thank you very much to draw our attention to this point and for your support on this manuscript.

We shortened the conclusion section as following:

“Aortic valve dysfunction is a rare, but relevant long-term complication after heart transplantation. TAVR is the preferred treatment in our center, but SAVR is feasible for cases ineligible for an interventional approach. The peri-procedural risks are higher than in non-transplanted patients, however 75% of the patients reported a subjective improvement in their well-being and functional status after the procedure.”

Reviewer #1 (Remarks to the Author):

The authors addressed the points I raised in reviewer comment.

I do not have any further comments.

Thank you very much for your support!

Reviewer #3 (Remarks to the Author):

Thank you for addressing my queries. I do not have any further issues with this manuscript.

Thank you very much for your support!